# Target Identification in Anti-Tuberculosis Drug Discovery

**DOI:** 10.3390/ijms241310482

**Published:** 2023-06-22

**Authors:** Rita Capela, Rita Félix, Marta Clariano, Diogo Nunes, Maria de Jesus Perry, Francisca Lopes

**Affiliations:** Instituto de Investigação do Medicamento (iMed.ULisboa), Faculdade de Farmácia, Universidade de Lisboa, Av. Prof. Gama Pinto, 1649-003 Lisboa, Portugal; ritacapela@ff.ulisboa.pt (R.C.); ritafelix@ff.ulisboa.pt (R.F.); martaclariano@campus.ul.pt (M.C.); dmnunes@edu.ulisboa.pt (D.N.); mjprocha@ff.ulisboa.pt (M.d.J.P.)

**Keywords:** *Mycobacterium tuberculosis*, target identification, activity-based probes, affinity-based probes

## Abstract

*Mycobacterium tuberculosis* (Mtb) is the etiological agent of tuberculosis (TB), a disease that, although preventable and curable, remains a global epidemic due to the emergence of resistance and a latent form responsible for a long period of treatment. Drug discovery in TB is a challenging task due to the heterogeneity of the disease, the emergence of resistance, and uncomplete knowledge of the pathophysiology of the disease. The limited permeability of the cell wall and the presence of multiple efflux pumps remain a major barrier to achieve effective intracellular drug accumulation. While the complete genome sequence of Mtb has been determined and several potential protein targets have been validated, the lack of adequate models for in vitro and in vivo studies is a limiting factor in TB drug discovery programs. In current therapeutic regimens, less than 0.5% of bacterial proteins are targeted during the biosynthesis of the cell wall and the energetic metabolism of two of the most important processes exploited for TB chemotherapeutics. This review provides an overview on the current challenges in TB drug discovery and emerging Mtb druggable proteins, and explains how chemical probes for protein profiling enabled the identification of new targets and biomarkers, paving the way to disruptive therapeutic regimens and diagnostic tools.

## 1. Introduction

*Mycobacterium tuberculosis* (Mtb), the etiological agent of tuberculosis (TB), is an obligate human pathogen spread by aerial transmission. One of the key mechanisms associated with Mtb virulence is the ability to subvert the host immune response, and to effectively avoid complete elimination via the host immune system [1,2,3]. In addition, the bacilli can co-exist within the host in a latent non-replicative form that is metabolically and physiologically different from the replicative state. Patients infected with latent Mtb bacilli are asymptomatic, and in low TB prevalence settings, most new active tuberculosis cases result from the reactivation of these pathogen forms [2,3,4].

Although TB is curable and preventable, with a treatment success of around 85%, the disease remains a global epidemic, estimated to be the second leading caused by a single infectious agent in 2021, only after COVID-19. Additionally, recent World Health Organization (WHO) data showed that the COVID-19 pandemic resulted into a rise in TB incidence, with a predicted observed maximum in 2022. Furthermore, around 25% of the world’s population is estimated to be infected, with 5 to 10% of those expected to develop active TB during their lifetime [5].

Most TB patients can be treated with currently approved drug regimens with reasonable efficiency, and in recent years, some novel drugs have been approved for the treatment of the more clinically challenging drug-resistant-TB (DR-TB), namely bedaquiline, delamanid, and linezolid. Nevertheless, there are still issues in anti-TB therapy that are yet to be addressed, though they should be in order to achieve TB control [5,6,7]. Most anti-TB drugs have limited efficiency against the latent bacilli, and therefore current therapeutic regimens require long durations to eliminate all forms of bacilli, usually leading to high adverse effects [8,9,10], poor compliance, and the emergence of drug resistance which consequently hampers TB control. Thus, there is an urgent unmet need for the development of anti-TB drugs that target latent bacilli and resistant strains [11].

A lack of knowledge about Mtb biology still limits the development of new diagnosis techniques and the conversion of new hit compounds into clinical candidates. Thus, there is a need to better understand the Mtb pathophysiology and find suitable biomarkers that allow the prediction of treatment responses and relapse risk, guarantee a cure, and accelerate drug development [6,7]. Currently, a diverse array of strategies are used to develop new anti-TB therapies. The most common include genetic approaches for the identification of new molecular targets, large-scale cell-based screening trials using Mtb, virtual screening, structural biology approaches, and the optimization of existent drugs through molecular modifications. Combinations between approaches based on validated targets and cell-based screening trials have gained attention in recent years and seem a promising strategy in discovering new active drugs [12,13,14].

After several decades without any novel anti-TB drugs being approved, major breakthroughs have been achieved in the last decade in the search for new therapeutic tools and regimens. Herein, we review the current challenges in TB drug discovery, discuss the emerging molecular targets that can leverage the discovery of new drugs, and address the development of chemical probes as a strategy to identify and validate novel targets in Mtb.

## 2. Challenges in TB Drug Discovery

TB drug discovery remains a challenging task due to the nature of its etiological agent, the heterogeneity of the disease, the emergence of resistance, and the lack of knowledge regarding the disease’s pathophysiology. One of the crucial requirements to achieve efficient treatment is the ability of a drug to enter into the target cell [15]. Compared to other bacteria, the Mtb cell wall is significantly less permeable to chemotherapeutical agents. Small hydrophobic molecules move quickly through the mycobacterial cell wall, while the movement of hydrophilic molecules is mediated by water-filled channels [16]. Moreover, when the bacteria are found intracellularly, a second permeability barrier exists, which further reduces the movement of drugs into the bacilli [17]. Additionally, the Mtb cell envelope also includes an array of efflux pumps which have an essential role in the physiology, metabolism, and cell signalling processes. These efflux pumps assist the expulsion of drugs from the mycobacteria and cause a natural high innate resistance to many anti-TB drugs [18]. 

Progression from latent infection to active TB constitutes a major source of active disease in developed nations, and it is becoming clear that tools to effectively address latency are needed to control the TB epidemy [4]. The Mtb latent state is characterized by a distinctive reduced metabolism, where ribosomal functions and aerobic respiration decrease and where lipid metabolism increases, with decreased permeability for hydrophilic molecules due to the thickened cell wall [19]. Thus, latent Mtb bacteria have an antibiotic tolerance, achieved by a combination of reduced antibiotic uptake and a lack of druggable targets as a result of the metabolic reconfiguration. Since anti-TB drugs specific and effective to latent bacteria are in short supply, current TB treatment is based on the prolonged administration of traditional anti-TB drugs. With the emergence of resistance and the pressure to shorten TB treatments, the ability to directly address latent subpopulations has become a priority in TB drug discovery, and the desire for agents that are capable of targeting all Mtb subpopulations has been emphasized [17]. Furthermore, the emergence of resistance has already led to the development of disease forms that are not treatable by any currently available therapeutical tool, such as total-drug-resistant TB (TDR-TB), which remains programmatically incurable [7,20]. Consequently, there is a pressing need to develop drugs that are active in unexplored targets and pathways and are not predisposed to resistance.

While historically effective, high-throughput screenings encounter several challenges in TB drug discovery. Despite the fact that they have general high hit rates, many compounds have undesirable physicochemical attributes, low selectivity, and mammalian cytotoxicity [17,21]. With the evolution of genomic tools, target-based screenings on validated drug targets, presumed to be indispensable for the survival of Mtb and pathogenicity, have gained some attention. A combination of ligand-based and structure-based chemogenomic approaches, followed by biophysical and biochemical validation, have also been used to identify targets for Mtb phenotypic hits [22]. However, this technique has several challenges as hits identified through target-based screenings may not translate to a whole cell system due to metabolic, permeability, and drug efflux issues. Since the publication of the entire Mtb genome sequence, and although several potential TB drug targets have been validated for use in target-based screening, no single clinically effective anti-TB agent has been discovered by this strategy [23,24]. 

Furthermore, the lack of predictive models for heterogeneous bacterial subpopulations is a limiting factor in TB drug discovery. To reproduce the environmental conditions of Mtb subpopulations, several in vitro models have been developed, such as hypoxia [25,26], nutrient starvation [27], low pH [28], multi-stress [29,30], and biofilm models [31], all with some limitations. Moreover, while very useful in early drug discovery stages, in vitro models cannot reproduce all host–pathogen interactions. Currently, the challenge of an adequate in vivo TB model remains, since existing animal TB models do not replicate important features of human disease [17,32]. An increased understanding of the microenvironments relevant to infection is difficult to achieve, but is urgently required to identify and validate new pharmacological targets and suitable biomarkers, and to consequently develop diagnostic techniques and improve therapy through several mechanisms such as the development of new pharmacological agents, the optimization of treatment durations, and the triage of high-risk patients to preventive treatment [33,34]. 

## 3. Emerging Mtb Drug Targets

The success of TB drug discovery requires the identification of compounds targeting proteins that are essential for the growth and survival of Mtb. Ideally, these molecular targets should also display low probability in order to undergo mutations and to prevent or delay the emergence of drug resistance [35]. While whole-genome sequencing has expanded our knowledge on Mtb cellular machinery, less than 0.5% of bacterial proteins are targeted in current therapeutic regimens [36]. Due to the development and spread of resistance to current drugs and the high toxicity associated with therapeutic regimens used in drug-resistant TB, there is an urgent need to discover new and safer drugs with novel mechanisms of action. The biosynthesis of the cell wall and the energetic metabolism of Mtb are critical cellular processes that are being exploited for TB chemotherapeutics.

### 3.1. Cell Wall

The cell wall of Mtb is the primary host–pathogen interaction spot, and is a major determinant of bacillus durability and robustness. The complex and dynamic structure of the cell wall (Figure 1) is essential for maintaining cellular integrity, enabling the adaptation of the bacilli to host conditions, and plays a crucial role in long-term infection and virulence. It comprises three essential substructures: a peptidoglycan (PG) inner layer, a mycolic acid (MA) outer layer, and an arabinogalactan polysaccharide (AGP) middle layer. The inhibition of key enzymes that are responsible for the biosynthesis of these substructures are excellent targets for novel drug development due to the absence of homologous characteristics in the host [37].

Peptidoglycan layer. The peptidoglycan is composed of *N*-acetylglucosamine (GlcNAc) and *N*-acetylmuramic acid (MurNAc) which are cross-linked with short peptides. The biosynthesis of peptidoglycan is a complex sequence of reactions, starting with the synthesis of lipid II, in which a hydrophobic polyisoprene tail embedded in the membrane is connected to a monomer of cell wall peptidoglycan through a pyrophosphate linker. This step is followed by the translocation of lipid II bound to the membrane formation, lipid II polymerization, and cross-linking with penicillin-binding proteins (PBPs) (including L,D-transpeptidases) [38]. Lipid II is targeted by the antibiotics ramoplanin and teixobactin, inhibiting the transglycosylation process and affecting peptidoglycan formation. 

Mtb also produces *β*-lactamase, an enzyme that catalyses the hydrolysis of *β*-lactam antibiotics, which explains why the use of these antibiotics is not included in TB treatment. However, carbapenems (Figure 2A) are resistant to inactivation with *β*-lactamases, and thus they are included in the treatment of multidrug-resistant TB, since they target the biosynthesis of peptidoglycan by inhibiting L,D-transpeptidases.

Mycolic acid layer. Mycolic acids are very long chain (C60-90) *α*-alkyl *β*-hydroxy fatty acids that contribute to the hydrophobic, impermeable, and rigid structure of the outer membrane [39,40]. Mycolic acids are synthesised from acetyl-CoA by at least two elongation systems, the type I and type II fatty acid synthases, also known as FAS-I and FAS-II. The FAS-II system can only be found in bacteria, turning this system into a potential selective antibacterial target. The NADH-dependent enzyme 2-*trans*-enoyl-acyl carrier protein reductase, InhA, is involved in the FAS-II system and is targeted by isoniazid (Figure 2B), a first-line agent used for treating TB. The mode of action of other small molecules, including ethionamide (Figure 2B), which is structurally related to isoniazid and triclosan, is related to InhA inhibition. 

Subsequent cycles of fatty acid elongation are carried out with *β*-ketoacyl synthase KasA, which completes chain elongation via the condensation of FAS-I-derived acyl-CoAs with malonyl-ACP (acyl carrier protein). KasA is the only essential member of three *β*-ketoacyl synthases encoded in the Mtb genome [41], and has been reported as a validated target for the treatment of TB [42]. A structure-based approach was used to optimize existing KasA inhibitor DG167 [43,44] to afford indazole JSF-3285 (Figure 2B) with a 30-fold increase in mouse plasma exposure. Biochemical, genetic, and X-ray studies further confirmed that JSF-3285 targets KasA. 

Mycolic acids are transported to the outer membrane due to bacterial membrane proteins called mycobacterial membrane protein large (MmpL), which are part of the resistance, nodulation, and cell division (RND) family. The primary role of RND proteins is to translocate a broad range of compounds across the plasma membrane to the periplasmic space, including virulence-associated envelope lipids and siderophores. The Mtb genome encodes 13 MmpL proteins, of which MmpL3 has been reported in the biosynthesis of the mycobacterial outer membrane. The ethylenediamine derivative SQ109 (Figure 2B) is a MmpL3 inhibitor and has completed phase IIb-III clinical trials. SQ109 also accumulates in the lungs, the site of infection, increasing the drug efficacy [6]. Other promising MmpL3 inhibitors include indolocarboxamides and adamantylureas. As part of a drug scaffold repurposing program, the cannabinoid receptor modulator rimonabant (Figure 2B) and its diaryl pyrazole analogs were reported to display potent anti-TB activity [45,46]. 

Arabinogalactan polysaccharide layer. The branched-chain arabinogalactan (AG) is the major cell wall polysaccharide, representing *ca* 35% of the cell wall, composed of arabinose and galactose residues, both in the furanose configuration. This middle layer is covalently attached to peptidoglycan and mycolic acid layers which require several enzymes that are potential targets for the design of novel inhibitors to block the formation of arabinogalactan polysaccharide [47,48,49,50], e.g., arabinosyltransferase enzymes (EmbA, EmbB, and EmbC), which are known targets for the drug ethambutol [51]. Another target used to block the arabinogalactan polysaccharide formation is the enzyme arabinofuranosyltransferase (Aft), responsible for the polymerization of arabinofuranyl residues in decaprenylphosphoryl-D-arabinose (DPA), the lipid donor of D-arabinofuranosyl residues of AG. The DPA synthetic pathway is a potential drug target, and several arabinosyltransferases are essential in the growth of Mtb, such as AftA, AftB, AftC, and AftD. While AftA and AftB are responsible for the transference of arabinofuranosyl residue, AftC and AftD introduce the α-1,3-branching in the segments of α-1,5-linked D-Araf residues [47,52]. 

The attachment of the arabinogalactan to the peptidoglycan structure is performed via an essential linker, the disaccharide L-rhamnose-D-*N*-acetylglucosamine. The enzyme *N*-acetylglucosamine-1-phosphate transferase, GlcNAc-1-P transferase (WecA), catalyses the first step of this linker biosynthesis. For this reason, WecA inhibitors, such as CPZEN-45 (Figure 2C), a caprazamycin derivative, prevent the growth of Mtb [53,54].

The enzymes decaprenylphosphoryl-*β*-D-ribose 2′-oxidase (DprE1) and decaprenylphosphoryl-D-2-keto erythropentose reductase (DprE2) are involved in the two-step epimerization of decaprenylphosphoryl-β-D-ribofuranose (DPR) into decaprenylphosphoryl-β-D-arabinofuranose [55]. Diverse chemical scaffolds such as azaindoles, aminoquinolones, benzothiazinones, benzothiazoles, dinitrobenzamides, nitrobenzamides, pyrazolopyridines, quinoxalines, triazoles, and thiadiazoles, demonstrated DprE1 inhibition. The benzothiazinone derivatives BTZ-043 and PBTZ169 (Figure 2C) are currently in phase II clinical trials and demonstrated high efficacy against *M. tuberculosis*. Additionally, the non-covalent inhibitors, azaindole TBA-7371 and OPC-167832, currently in phase II and phases I/II clinical trials, respectively, have shown promising results [56].

### 3.2. Energy Metabolism 

Mtb operates its energetic metabolism in a modular and compartmentalized mode to support distinct and key cellular functions [2,19]. 

Electron Transport Chain. Mtb relies on oxidative phosphorylation (OxPhos) via the electron transport chain (ETC) to produce energy for growth and division purposes. During the OxPhos process, electrons are transferred from electron donors produced in the central metabolic pathways to molecular oxygen through the ETC. The energy released in this process is conserved by proton-pumping transmembrane proteins that establish a proton gradient and thus generate an electrochemical gradient, called proton motive force (PMF). This bioenergetic pathway generates ATP from the phosphorylation of ADP [57,58]. 

The Mtb ETC is a highly conserved collection of membrane-bound and membrane-associated enzymes and co-factors. It is comprised by five main primary dehydrogenases, which fuel the ETC as electron donors; two main terminal oxidoreductases, which catalyse the transfer of electrons to terminal electron acceptor; and an ATPsynthase, which produces ATP through the dissipation of the PMF. A schematic representation of the Mtb ETC is presented in Figure 3 [59,60,61].

The respiratory flexibility of Mtb, which allows the bacilli to vary the ETC enzyme composition in response to environmental conditions, as well as the existence of human homologs to most ETC enzymes, hampered the development of selective inhibitors. However, the discovery of bedaquiline, an ATPsynthase inhibitor, leads to an increase in research focused on targeting OxPhos. Currently, more than 30% of all new antimycobacterial drugs in clinical trials target the OxPhos, and more than 65% of phase III trial regimens include a OxPhos inhibitor [59].

Cytochrome bc_1_-aa_3_. The Mtb Cyt bc_1_-aa_3_ supercomplex comprises two tightly associated protein complexes: a menaquinol–cytochrome c oxidoreductase (or cyt bc_1_) and an aa_3_ oxidase (or cyt aa_3_)_._ This supercomplex acts as the primary terminal oxidase under normoxia, and, during exponential growth, its inhibition results in growth arrest. However, cyt bc_1_-aa_3_ is not essential for cell survival and as long as the alternate cyt bd is expressed, bc_1_-aa_3_ inhibitors do not induce bactericidal effects. The central role of the bc_1_-aa_3_ complex in the ETC and the significant differences to the mammalian counterpart make the supercomplex a good therapeutical target [58,59,60,61,62,63,64,65,66].

Imidazopyridine derivatives are examples of inhibitors that have shown to be particularly potent, the most prominent example being Q203, which is currently in phase II clinical trials and is capable of inhibiting multidrug-resistant (MDR) TB and extensively drug resistant (XDR) Mtb strains [59,67,68,69]. Structurally similar to Q203, TB-47 has been reported in pre-clinical studies and is active against drug-sensitive (DS) and drug-resistant (DR) Mtb strains, including both active and latent bacilli [70,71]. Lansoprazole, a gastric proton pump inhibitor, was found to be a potent hit compound in the screening of FDA-approved drugs. Lansoprazole acts as prodrug and is converted in vivo into lansoprazole sulphide, which was identified to be a cyt bc_1_-aa_3_ inhibitor on a distinct site from the one targeted by imidazopyridines (Figure 4A) [61,66,72].

Cytochrome bd. Cytochrome bd-type menaquinol (MKH_2_) oxidase, or cyt bd, is a non-proton pumping, and is a less energetically efficient terminal oxidase that transfers electrons from MKH_2_ to molecular oxygen. Cyt bd is exclusive to the prokaryotic ETC, and, unlike cyt bc_1_-aa_3_, the enzyme is more versatile, with multiple functions reported. The terminal oxidase is capable of detoxifying ROS and antibacterials, and protects the bacilli against hypoxia, and is capable of compensating the inactivation of cyt bc_1_-aa_3_. Cyt bd may also play a role into the Mtb’s natural drug tolerance, namely to drugs that directly target the ETC [59,60,73,74]. Thus, this cytochrome contributes to Mtb virulence, and since the enzyme is not encoded in animal genomes, it can serve as an attractive promising therapeutical target for new selective anti-TB drugs [67,72,75].

The inhibition of cyt bd alone does not have any antimycobacterial effects. However, cyt bd inhibitors have synergetic effects with isoniazid, quicken the bactericidal activity of ATPsynthase inhibitors, and turn bc_1_-aa_3_ inhibitors bactericidal [59,60,61]. Thus, cyt bd inhibitors appear to be particularly attractive in combination therapy, namely in combination with cyt bc_1_-aa_3_ inhibitors, as the simultaneous inhibition of both terminal oxidases is highly bactericidal in a short period of time and is successful at killing both active and latent bacilli. The non-essentiality of cyt bd represents a challenge in order to identify its inhibitors, and thus not many cyt bd inhibitors are known. To this date, only a few were identified and only one, aurachin D (Figure 4B), is characterized. Aurachins are isoprenoid quinoline alkaloids, originally extracted from myxobacteria. The further development of aurachin D is complicated by its toxicity and a lack of selectivity, but optimized derivatives of aurachin D have great potential as anti-TB drugs [67,72,74,75,76].

Delamanid (DLM) and pretomanid (PTM) (Figure 4B) are two structurally related nitroimidazoles that were recently approved for the treatment of MDR-TB and were found to inhibit the biosynthesis of mycolic acid. However, the observation that these drugs were bactericidal against both active and latent bacilli suggested an alternative mechanism of action, as mycolic acid biosynthesis is downregulated in latency. Both DLM and PTM are pro-drugs that require activation with an F420 nitro-reductase, an enzyme which produces des-nitro metabolites with the release of NO. The putative additional mechanism of action is that the intracellular release of NO poisons the cytochrome oxidases, resulting in respiration arrest and consequent cell death. DLM and PTM treatment results in a quick decrease in intracellular ATP levels, an increased menaquinol–menaquinone (MKH_2_/MK) ratio, and the upregulation of cyt bd and nitrate reductase, which further support the concept of terminal oxidases being used as targets of these nitroimidazoles [59,60,69]. 

ATPsynthase. Bedaquiline (BDQ, Figure 4C), an inhibitor of the ATPsynthase approved by the FDA in 2012, was the first drug specifically approved for TB in more than 40 years. BDQ is a potent bactericidal efficacious against MDR and latent bacilli, and currently is conditionally administrated for MDR-TB treatment [59,68,77]. The proposed mode of action is to bind to two subunits on the ATPsynthase, thus inhibiting ATP synthesis and leading to a depletion of intracellular ATP levels. Additionally, bedaquiline is capable of acting as a protonophore, leading to the uncoupling of the ETC via the collapse of the PMF. Inhibition with BDQ depletes intracellular ATP levels; activates respiration; and induces a metabolic remodelling that upregulates ATPsynthase, NDH-2, and cyt bd. Interestingly, the bacterial activity of BDQ is delayed, i.e., it does not occur immediately upon the ATP depletion, as explained by the metabolic remodelling Mtb experiences upon BDQ exposure [57,59,67,77].

The toxicity associated with drugs and the emergence of bedaquiline-resistant Mtb strains restrain its use to MDR- and XDR-TB patients. Thus, in order to address its shortcomings, a medicinal chemistry approach was conducted to study the chemical space of diarylquinolines to find next-generation equivalents with superior safety profiles. In this context, two 3,5-diakoxy-4-pyridyl derivatives, TBAJ-587 and TBAJ-876 (Figure 4C), were found to be particularly interesting and are currently in phase I clinical trials [58,61,68,77]. 

A number of recent studies have identified new ATPsynthase inhibitors with novel mechanisms of action. A family of squaramide derivatives was found to be particularly interesting, with its lead compound (Figure 4C) currently being evaluated in pre-clinical trials. These compounds target ATP synthase through a different binding site, meaning that they do not show cross-resistance to BDQ, and have shown to be active against BDQ-resistant strains [57,59,67].

Other targets on the ETC. The PMF consists of an electrical potential due to charge separation across the membrane and the chemical potential of protons. The generation and maintenance of a PMF is essential for Mtb energy production and consequent bacterial growth and survival in every metabolic state. PMF uncouplers generally act as protonophores and uncouple OxPhos from the ETC, thus inhibiting ATP synthesis, leading to cell death [78]. Generally, this kind of compounds is not sufficiently selective to be used as antimycobacterial agents, and thus the development of specific PMF uncouplers remains an area of interest. However, there are some examples of anti-TB drugs in clinical use that act as PMF uncouplers in addition to an alternative mode of action, such as bedaquiline (Figure 4C), pyrazinamide (PZA), nitazoxanide (NTZ) (Figure 5), and SQ109 (Figure 2B) [57,78]. 

Although the mechanism of action of PZA is not still fully understood, current knowledge indicates that it acts as a multitarget drug that dissipates the PMF, inhibits ATP synthesis, inhibits membrane transport, and reduces the activity of other proteins (such as aspartate decarboxylase, a protein involved in the coenzyme A biosynthetic pathway). Evidence of PZA’s uncoupling activity first arose with its ability to target latent bacilli. Additionally, PZA showed to synergize with other PMF uncouplers to deplete ATP depletion and enhance mycobacterial killing, implying that its anti-TB activity substantially relies on its uncoupling activity [57,61]. 

Initially, SQ109 was reported to interfere with the assembly of mycolic acids in the mycobacterial cell wall through the inhibition of membrane transporter MmpL3, but recently, it was demonstrated that SQ109 interferes with respiration due to their ability to act as a protonophore and dissipate the PMF [59,68]. 

Nitazoxanide is an FDA-approved repurposed drug with broad-spectrum antiparasitic and antiviral activity. NTZ is proposed to promote Mtb killing by enhancing autophagy through the inhibition of human mTORC1 and disrupt the PMF by acting as a protonophore. NTZ potently inhibits both active and latent Mtb, bacilli but has poor pharmacokinetic and pharmacodynamic proprieties. Thus, there is some interest in the development of NTZ derivatives with improved bioavailability [59,79,80]. 

### 3.3. Other Targets 

#### 3.3.1. Iron Uptake

Iron is fundamental in Mtb survival, and, for this reason, all the systems involved in iron uptake are promising drug targets [81]. When in unfavourable iron-deficient environments, Mtb increases the uptake of iron through the synthesis of high-affinity iron chelators, called siderophores. Targeting the biosynthesis of mycobactin siderophores from mycobacteria has been exploited as an approach to inhibit the growth of Mtb. The Mg^2+^-dependent salicylate synthase (MtbI) enzyme is a validated target since it is responsible for salicylate synthesis from chorismate in the first step of the mycobactin biosynthesis pathway. Furthermore, MtbI offers the potential to enable the discovery of highly selective inhibitors, as it is absent in the host. Using a receptor-based virtual screening procedure, several furan-based compounds (Figure 6) were identified as potent MtbI inhibitors [82,83].

Another approach to block iron uptake in Mtb is to inhibit the iron-dependent transcription factor, IdeR, which controls siderophore synthesis. This regulator is a DNA-binding protein of the DtxR family that is responsible for the activation or deactivation of storage proteins, according to the excess or lack of iron, respectively. A screening against IdeR revealed benzothiazole benzene sulfonic (Figure 6) as a promising scaffold to develop IdeR inhibitors [6,84,85].

#### 3.3.2. DNA-Related Enzymes

DNA gyrase. This enzyme is a validated target for anti-tubercular drug discovery. It is an ATP-dependent enzyme that is essential for efficient DNA replication, transcription, and recombination in bacteria [86]. Moreover, its absence in the mammalian organism makes this enzyme a suitable target for the development of antibacterial drugs with selective toxicity. Fluoroquinolones are effective inhibitors of this enzyme (Figure 7A). As with other antitubercular drugs, side effects and emerging bacterial resistance have fuelled intensive research for new chemical entities, from natural or synthetic origin, possessing DNA gyrase inhibiting properties that would be effective against MDR-TB, and could also be effective against fluoroquinolone-resistant Mtb [87].

DNA Topoisomerase I. Imipramine and norclomipramine (Figure 7B) showed the growth inhibition of both *Mycobacterium smegmatis* and Mtb cells. They target DNA topoisomerase I, an essential mycobacterial enzyme in the maintenance of topological homeostasis within the cell, during a variety of DNA transaction processes such as replication, transcription, and chromosome segregation. It was suggested that they bind near the metal-binding site of the enzyme, so targeting metal coordination in topoisomerases may be a general strategy used to develop new lead molecules [88].

DNA ligases. Vital enzymes in replication and repair, DNA ligase catalyses the formation of a phosphodiester linkage between adjacent termini in double-stranded DNA through similar mechanisms. The DNA ligases either utilize ATP or NAD^+^ as cofactors. Those utilizing NAD^+^ are attractive drug targets because of the unique cofactor requirement for ligase activity and are exclusively found in eubacteria and some viruses. Gene knockout and other studies have shown NAD^+^-dependent DNA ligases to be indispensable in several bacteria (including Mtb). Compounds belonging to arylamino and pyridochromanone classes (Figure 7C) have been identified as specific inhibitors of NAD^+^-dependent DNA ligases and can potentially be used to develop novel antibacterial therapies [89,90].

Thymidine kinase. Thymidine monophosphate kinase (TMPK) catalyses the *γ*-phosphate transfer from ATP to thymidine monophosphate (dTMP) in the presence of Mg^2+^, yielding thymidine diphosphate (dTDP) and ADP. Because TMPK is essential for thymidine triphosphate (dTTP) synthesis, and in the view of its low sequence identity (22%) with the human isozyme (TMPKh) and its unique catalytic mechanism, it represents an attractive target for selectively inhibiting mycobacterial DNA synthesis [91]. Both industrial and academic efforts have afforded several potent Mtb TMPK inhibitors in the last two decades, including thymidine-like and non-nucleoside inhibitors (Figure 7D) [91].

## 4. Chemical Probes for Target Identification in Mycobacteria

The search for new biomarkers and potential drug targets in Mtb has led to the development of chemical probes as tools for protein profiling proteomic methodologies. Activity-based protein profiling (ABPP) is a proteomic technique that enables the quantification and functional analysis of enzymes using activity-based probes (ABPs). Typically, ABPs react covalently with the active form of an enzyme or mechanistically related classes of enzymes. ABPs include (i) a reactive group (or warhead) that reacts with the catalytic amino acid residue of the enzyme, (ii) a reporter tag (e.g., a biotin for protein pulldown or a fluorophore for cell imaging), and (iii) a linker bridging the warhead and the tag. 

In contrast to ABPP, profiling non-catalytic proteins relies on the use of photoaffinity-based probes (A*f*ABPs) that incorporate a photo-activable moiety to enable the covalent crosslinking between the probe and the target protein upon irradiation [92]. Concerning the reporter tag, a biorthogonal handle can be used instead of the biotin or the fluorophore. In this strategy, after the linkage of the probe to the target, a copper alkyne-azide cyclization is performed in the living cell or in cell lysates to incorporate the tag for analysis (Figure 8).

### 4.1. Activity-Based Protein Profiling (ABPP)

ABPP studies have been used to explain biological pathways in *Mycobacterium tuberculosis*, find new therapeutic targets, or identify new biomarkers.

#### 4.1.1. Cytosolic Serine Hydrolases

Several serine hydrolases have been reported as putative targets in bacterial infections [93,94], and, in the case of Mtb, represent 1.2% of all proteomes [95]. Ortega et al. reported an extensive ABPP study in replicating and non-replicating Mtb in order to identify serine hydrolases that remain active in non-replicating persistent states. Using a pan-serine hydrolase fluorophosphonate probe, FP-ABP (Figure 9), 78 hydrolases of a total of 208 proteins were identified. The activity of these 78 hydrolases was analysed in normoxia and hypoxia and only 3 were active in hypoxic conditions, while 41 were active in aerated cultures and 34 were active in both conditions. Overall, these data provided experimental validation for previously annotated Mtb enzymes and identified 37 FP-labelled proteins that were found to be active in non-replicating Mtb that could be used as new drug targets for persistent Mtb. Specifically, mycobacterial acid resistance protease (MarP) activity was shown to remain unchanged between both phenotypes, suggesting a role in maintaining persistence. The ClpP2 subunit, included in the Mtb ClpP protease complex, was shown to be the only one detected in non-replicating Mtb [96,97].

Lentz et al. designed activity-based probes (ABPs) to selectively target Mtb “Hydrolase important to pathogenesis” (Hip 1). This is a cell-envelope-associated serine protease whose proteolytic activity is required for the immunomodulation of host inflammatory responses and has weak homology to other host proteases. From a library of serine-reactive electrophiles, a series of 7-amino-4-chloro-3-(2-bromoethoxy)isocoumarins were identified as potent time-dependent inhibitors of Hip1, and were used to synthesise the fluorescence probe (Figure 10A). While this ABP displayed high potency but low selectivity, optimizing the isocoumarin scaffold led to the isocoumarin (Figure 10B), an inhibitor with nanomolar activity against Hip1 and improved selectivity [98].

Isocoumarins were also identified as inhibitors of Mtb growth in a screening of a library containing electrophiles that react with serine hydrolases. In particular, the seven-urea chloroisocoumarin JCP276 was active against Mtb and *Mycobacterium kansasii*, but had no effect against other non-tuberculous mycobacteria. A competitive gel-based ABPP assay with fluorophosphonate-tetramethylrhodamine (FP-TMR) and JCP276 showed that the hit compound was able to interact with several proteins. The proteomic ABPP study using the BMB034 probe allowed seven major targets to be identified, mostly between lipases and esterases (Figure 11). However, the inhibition of these targets individually did not affect cell growth, which suggests that the potency of JCP276 may arise from the inhibition of multiple targets [99].

In another study, by combining the usual tools of ABPP with a competitive approach, Li et al. screened a 1,2,3-triazole urea library of ca. 200 molecules with the aim of identifying the serine hydrolase that are implicated in the Mtb growth. First, using a fluorophosphonate biotin probe, FP-biotin (Figure 12), several serine hydrolase targets were revealed, and the selected targets were then tested for the two most active 1,2,3-triazole ureas. The results showed that the antimycobacterial activity displayed by these compounds is related to the inhibition of several key serine hydrolases that are essential in lipid metabolism and cell wall biosynthesis. The competitive ABPP study with FP-TMR showed the multiple target inhibition that led to cell wall disruption and lipid metabolism [100].

#### 4.1.2. Membrane Serine and Cysteine Hydrolases

ABPP also proved to be instrumental in revealing the key role of serine hydrolases in mycobacterial cell wall biosynthesis, leading to the identification of potential inhibitors of these enzymes. *β*-Lactam antibiotics interferes with the final phase of the biosynthesis of peptidoglycan by inhibiting irreversible D,D-transpeptidase serine hydrolases, known as penicillin-binding proteins (PBPs), thus preventing the formation of bonds between peptide chains of peptidoglycan (“cross-linking”). However, in Mtb, the peptidoglycan layer contains different peptide crosslinks which mostly require catalysis with L,D-transpeptidases, i.e., cysteine hydrolases [101,102].

The presence of β-lactamase in TB bacilli raises the question of whether this group of antibiotics can be used against Mtb. However, the combination of certain *β*-lactams, carbapenem, and meropenem, along with a *β*-lactamase inhibitor used as clavulanic acid, revealed an antitubercular activity improvement, which reiterates the importance of *β*-lactams in TB treatment. 

Quezada et al. screened a library of *β*-lactams against Mtb under replicating and non-replicating conditions and found two cephalosporins, exclusively active against non-replicating Mtb. To explore the possibility of an alternative and less known pathway beyond the action of transpeptidases, chemical probes were designed to perform ABPP studies (Figure 13) [103].

With a specific focus on the relevance of L,D-transpeptidases for cell wall biosynthesis, Munnick et al. developed an assay based on the use of cysteine-selective fluorogenic probes for testing the reactivity with L,D-transpeptidases, which appears to be of special importance for Mt virulence. In this assay, two fluorogenic probes based on benzoxadiazole and fluorescein were tested in the presence of competitive inhibitors for L,D-transpeptidases, as well as several *β*-lactams antibiotics (Figure 14). This study revealed penems and carbapenems to be potent inhibitors of L,D-transpeptidases [104].

Similar to other bacterial infections, penicillin and cephalosporin *β*-lactam antibiotics fail in TB therapeutics due to inactivation using *β*-lactamases. However, based on the observation that carbapenems can reduce the activity of this enzyme, Levine et al. developed activity-based probes based on a carbapenem derivative meropenem, Mero-Cy5 (Figure 15), with the aim of finding a class of enzymes and the mechanism of action of meropenem. This probe inhibited L,D-transpeptidases as it binds to an active-site cysteine residue, but also binds to other transpeptidases and carboxypeptidases, as well as to *β*-lactamase. The probe designed by Levine et al. proved to be a powerful tool for target identification and highlights the potential of carbapenem *β*-lactam antibiotics to treat TB [105]. 

Serine hydrolases play a crucial role in catalysing essential transacylation reactions, namely in binding mycolates as *β*–keto or hydroxyesters. The similarity between the *β*-lactone pattern and mycolates enables the covalent acylation of catalytic serine residues with *β*-lactones. Based on this, Lehmann et al. developed *β*-lactones that could covalently inhibit these enzymes, preventing the formation of a mycobacterium membrane. A *β*-lactone developed by this group exhibited good activity and selectivity for mycobacteria and they synthesised an alkyne probe, EZ120P, to identify the molecular targets (Figure 16). Through standard ABPP procedures, the serine proteases Pks13 and Ag85, essential proteins in the biosynthesis of the mycobacterial cell wall, were identified as possible targets [106].

Tetrahydrolipstatin (THL) is an inhibitor of fatty acid synthetase, an enzyme that plays an important role in latent tuberculosis, since fatty acids are crucial for the survival of mycobacterium in this phase. This inhibitor contains a *β*-lactone group which forms covalent adducts with serine residues of target enzymes. Ravidran et al. synthesised a THL-ABP (Figure 17), and, using click chemistry, the fluorescent-tagged THL proteins allowed their targets in mycobacteria to be determined. From 14 possible targets (*α*/*β*-hydrolases, including many lipid esterases), 2 of them were validated through several experimental techniques, lipH and tesA, which are fundamental lipolytic enzymes in the dormant state of mycobacteria [107].

Tallman et al. developed probes to identify Mtb esterases in active and dormant conditions, leading to the discovery of several esterases involved in the different states of the Mtb. With a red fluorescent TAMRA-FP probe, serine hydrolases were detected in replicating dormant and reactivation conditions, but their enzymatic activity was reduced in dormancy. However, using ABPP (desthiobiotin-fluorophosphonate) and fluorogenic (DCF-AME) probe-based profiling, it was possible to identify esterases present in dormant conditions (Culp1, LipH, LipM, LipN, and Rv3036c) or in both states (CaeA, Rv0183, and Rv1683) (Figure 18) [108,109,110].

#### 4.1.3. Other Membrane Targets 

*Acyltrehaloses biosynthesis.* Acyltrehaloses are components of the outer membrane of Mtb and have been identified as antigens, and have drawn interest as diagnostic markers with the potential to distinguish between tuberculous and nontuberculous mycobacteria [111].

Polyacyltrehalose (PAT) is the predominant acyltrehalose in Mtb, and, together with diacyltrehalose (DAT), has a structural function in the cell envelope and plays a role in the Mtb’s ability to replicate and persist in the host by facilitating Mtb intracellular survival and modulating host immune responses [112]. The enzymes involved in PAT biosynthesis have not yet been identified. The PAT biosynthetic gene locus is identical to that of sulfolipid 1, a trehalose glycolipid structurally analogous to PAT, which is also unique to virulent Mtb. Chp1, a cutinase-like hydrolase protein, was already described as the terminal acyltransferase in sulfolipid 1 biosynthesis [113]. Chp2 may play a role in the biosynthesis of PAT, which is reinforced by the coordinate upregulation of the *chp2* gene, a homologue of *chp1* (*rv3822*). However, the specific role of Chp2 is still not confirmed. 

Touchette et al. hypothesized that Chp2 is an acyltransferase responsible for the transformation of DAT in PAT, once the PAT biosynthetic gene cluster includes *chp2* (*rv1184c*). To confirm the enzymatic activity of Chp2, the authors have resorted to an activity-based probe FP-TMR, (Figure 11B), consisting in a fluorescent labelling reagent that specifically modifies the active-site residue of serine hydrolases. Thus, it was verified that Chp2 contains a C-terminal serine hydrolase domain that is inhibited by the lipase inhibitor tetrahydrolipstatin (THL). Results have also shown that THL inhibits Chp2, leading to decreased levels of PAT and the accumulation of DAT, suggesting that Chp2 is responsible for the synthesis of PAT from DAT, and plays an analogous role to the Chp1 in sulfolypid 1 byosynthesis [111].

*Fatty acid biosynthesis.* Knowing that fatty acids play an important role in mycobacteria, Ishikawa et al. developed probes with the aim of identifying dehydratase (DH) enzymes in fatty acid synthases (FASs). These probes contain a specific reactive sulfonyl-3-alkyne warhead to prevent hydrolysis or non-enzymatic inactivation. The designed probes in Figure 19 were based on the 3-decynoyl-*N*-acetylcysteamine [3-decynoyl-NAC] structure, a known inhibitor of dehydratase FabA and an important enzyme in fatty acid biosynthesis mechanism. The different experiments performed led to the following conclusions: (i) these fluorescent probes are selective inhibitors for dehydratase enzymes in FASs, (ii) the sulfonyl alkyne scaffold is required for stability, (iii) the probes exhibit antibiotic activity, and (iv) DH-containing enzymes are identified and selectively isolated [114].

*Sulfomucins degradation.* Mucins, including sulfomucins, are a type of mucosal fluid with antimicrobial properties and play a crucial role in protecting the host from the invasion of pathogens, forming a physical barrier with direct antimicrobial activity. Mucin degradation with bacteria is often regarded as an initial stage in pathogenesis since it can disturb the protection of host mucosal surfaces [115,116]. In mycobacteria, sulfatases are responsible for sulfomucin degradation and thus play a role in the pathogenicity of mycobacteria and in the hydrolysis of the *N*-sulfate group in sulfated glycosaminoglycans, thereby modulating bacterial adhesion [117,118]. To detect sulfatase activity in mycobacteria, Yoon et al. developed an activity-based probe that forms a *N*-methyl isoindole compound after intramolecular cyclization via the action of sulfatase enzyme, responsible for a coloured precipitate (Figure 20). It was verified in cultures of *Mycobacterium avium* and *Mycobacterium smegmatis* that the probe gave rise to a coloured precipitate after cleavage, indicating that this probe can be very useful in the detection of sulfatase activity in Mtb, presenting the advantage of being detected by the naked eye [119].

*Mycobactins biosynthesis.* Iron is fundamental for Mtb survival, and, to compensate the lack of iron, the bacteria developed mycobactins, a type of iron-chelating molecule, also called siderophores, responsible for shuttle-free extracellular iron ions entering the cytoplasm of mycobacterial cells.

Since the adenylating enzyme MbtA is crucial in the biosynthesis of mycobactins, Duckworth’s group developed the probe Sal-AMS ABP based on a potent selective inhibitor of MbtA (Sal-AMS) (Figure 21), with benzophenone as the photoreactive group and a small alkyne as the reporter group. The assays performed with this probe demonstrated an extraordinary specificity for MbtA in crude mixtures with other enzymes and the possibility of identifying adenylating enzymes in other organisms, such as *E. Coli* [120].

#### 4.1.4. ATP-Binding Enzymes

ATP-binding proteome. Wolfe et al. developed a different approach that applies activity-based chemoproteomic experiments to selectively profile the ATP-binding proteome in normally growing and hypoxic Mbt. The study was carried out in the Mtb H37Rv strain and used a desthiobiotin-conjugated ATP as a molecular probe (Figure 22), where the desired enzymes are covalently modified with biotin, and, after a pull-down, the target proteins can be identified through chemoproteomic experiments. This chemoproteomic technique may be used to broaden the functional annotations and physiological roles of many nucleotide-binding proteins and supports the evidence on the potential of antimicrobial inhibitors whose mode of action relies on competition within the ATP-binding site of select proteins. With this approach using an enriched subproteome of desthiobiotin-labelled ATP-binding proteins (ATPome), 176 proteins were identified in total, of which 122 were labelled via the molecular probe, and more than half have been reported to be essential for in vitro survival [121].

### 4.2. Affinity-Based Probes (AfBPs)

The biosynthesis of mycolic acid is carried out using two fatty acid synthases, FAS-I and FAS-II, where FAS-II is responsible for the synthesis of very long acyl chains. The mycobacterial FAS-II system works via the interaction between the acyl carrier protein (AcpM), which binds the growing acyl chain, and its respective enzymes, such as ketosynthases (KasA/KasB), reductases (MabA), dehydratases (HadAB/HadBC), and enoyl reductase (InhA), which are responsible for further processing [122,123].

Thioacetazone (TAC) is a bacteriostatic anti-TB drug whose use was restricted due to severe side effects and the frequent emergence of resistance. TAC’s mechanism of action was not confirmed, but it was presumed that it interferes with mycolic acid biosynthesis, based on the isolation of several truncated hydroxy fatty acids, leading to the suggestion that dehydratases in the FAS-II system could be possible targets. Moreover, there is a hypothesis that monooxygenase EthA activates TAC that then binds to the dehydratase complex HadAB through a cysteine residue (Cys61) (Figure 23). Another potential target is the dehydratase enzyme HadC in the complex HadBC, but there is no evidence of that [124].

Singh et al. developed a TAC-affinity-based probe (Figure 24) with azidonaphthalimido butanoic acid used as a fluorescent template to establish the target enzymes of the drug. The results showed the formation of cross-links with the HadAB complex in the presence or absence of the monooxygenase EthA, indicating that the HadAB complex has an affinity for TAC itself. Furthermore, it was observed that the HadA component in HadAB and the HadC component in the HadBC are targets of TAC or its oxidized forms. The selectivity of this probe towards the dehydratases HadAB and HadBC was also verified since the other dehydratase present in Mtb was not found to be a primary target. Additionally, this probe has the advantage of promoting cross-linking with the target protein under white light exposure, contrasting to UV-activated photo-affinity probes, which can lead to protein degradation [125].

Trehalose dimycolates (TDMs) are the most abundant glycolipids in Mtb’s cell wall and play an important role in Mtb’s pathogenesis regarding protection against severe environmental conditions. TDMs also have immunomodulatory functions, including the prevention of phagosome–lysosome fusion, allowing the bacteria to survive inside the host macrophage. Additionally, TDMs have a crucial role in granuloma formation. However, despite the importance of TDM, only one receptor is known, the macrophage inducible C-type lectin (mincle) [126].

Khan et al. synthesised an affinity-based proteome profiling (A*f*BPP) TDM probe (Figure 25), formed by a benzophenone group as a photoreactive trap and by an alkyne tag. The reactive carbohydrate portion is unfunctionalized and has a small and hydrophobic trap and tag system, making it a good mimic of the original TDM. The authors reported that this probe was then validated to be used for proteomic studies, since it can activate macrophages and appears to be a suitable TDM mimic [127].

## 5. Conclusions

Tuberculosis remains a global epidemic threat due to the complexity of the disease and the emergence of drug resistance. New breakthroughs and insights into the development of safer and more efficient drugs require a more comprehensive explanation of Mtb drug targets associated with TB pathophysiology. Many of the emerging targets with potential to positively impact anti-TB drug discovery are involved in cell wall synthesis, energy metabolism, iron uptake, and DNA synthesis. The development of activity-based and photoaffinity-based methodologies, combined with the most recent developments in proteomic methodologies, provided the TB scientific community with powerful tools to identify novel molecular targets and shed some light on the understanding of biological pathways in Mtb. The full integration of classical phenotypic screening and genomic approaches with proteomic-based protein profiling will be instrumental in identifying effective targets to develop new safer and efficacious drug candidates that are capable of addressing the current challenges in TB therapy.

## Figures and Tables

**Figure 1 ijms-24-10482-f001:**
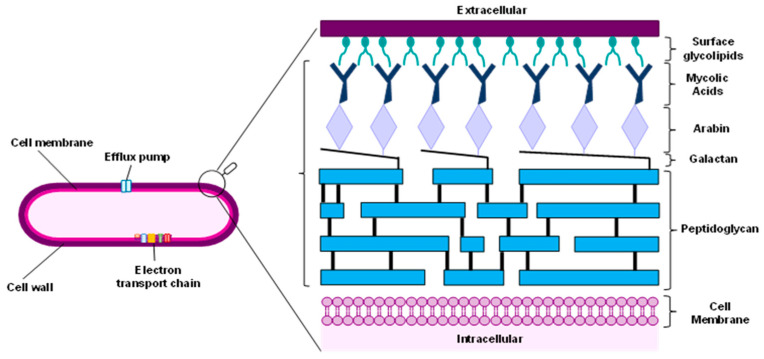
Composition of the cell wall of mycobacterium tuberculosis.

**Figure 2 ijms-24-10482-f002:**
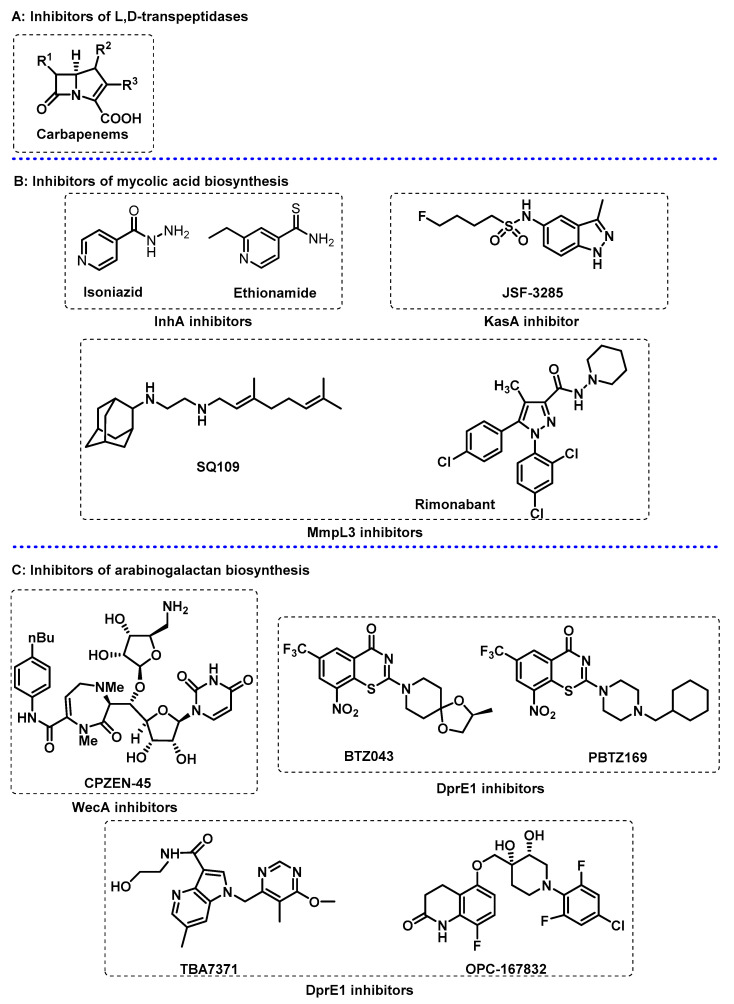
Selected compounds that target the cell envelope of *Mycobacterium tuberculosis*. (**A**) Carbapenems as inhibitors of L,D-transpeptidases; (**B**) inhibitors of the biosynthesis of mycolic acids; (**C**) inhibitors of the biosynthesis of arabinogalactan.

**Figure 3 ijms-24-10482-f003:**
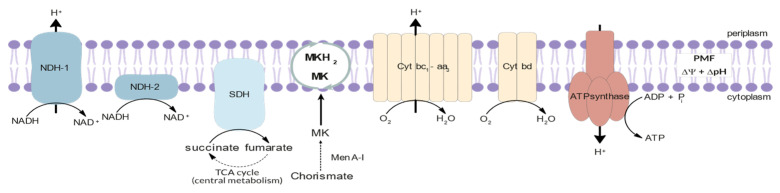
Schematic representation of the electron transport chain of *Mycobacterium tuberculosis*.

**Figure 4 ijms-24-10482-f004:**
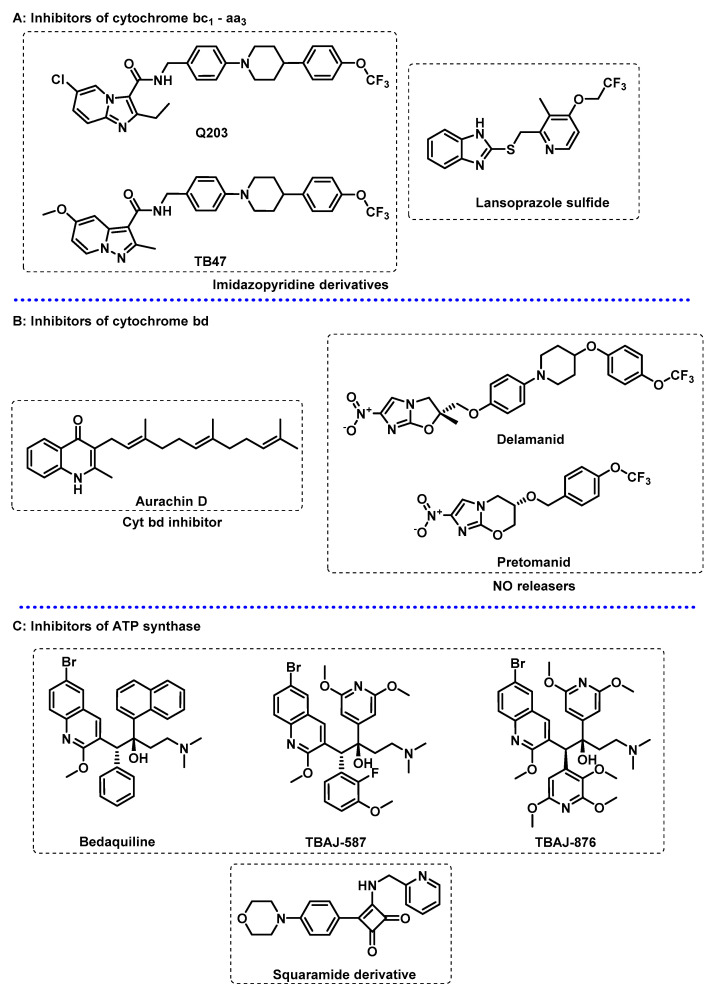
Selected compounds that target the energy metabolism of *Mycobacterium tuberculosis*. (**A**) Inhibitors of cytochrome bc_1_-aa_3_; (**B**) inhibitors of cytochrome bd; (**C**) inhibitors of ATP synthase.

**Figure 5 ijms-24-10482-f005:**
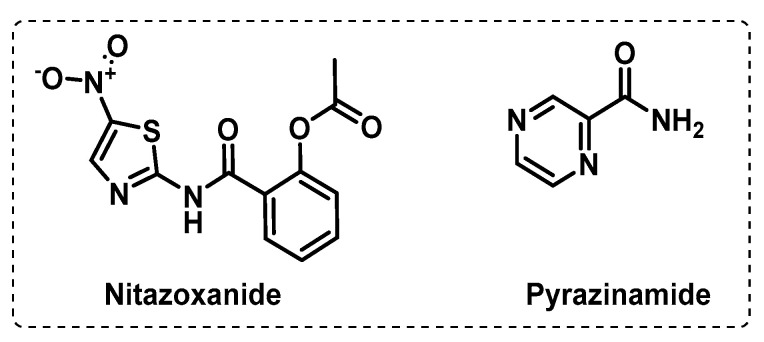
Proton motive force uncouplers in current clinical settings.

**Figure 6 ijms-24-10482-f006:**
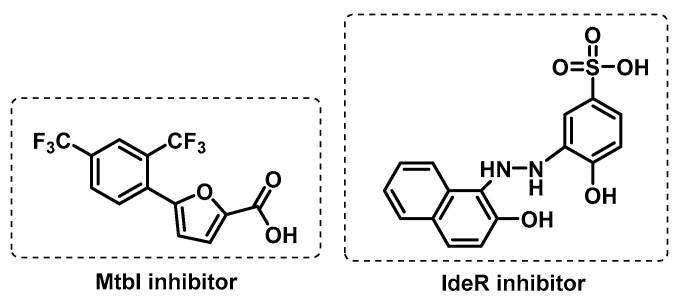
Example of iron uptake inhibitors.

**Figure 7 ijms-24-10482-f007:**
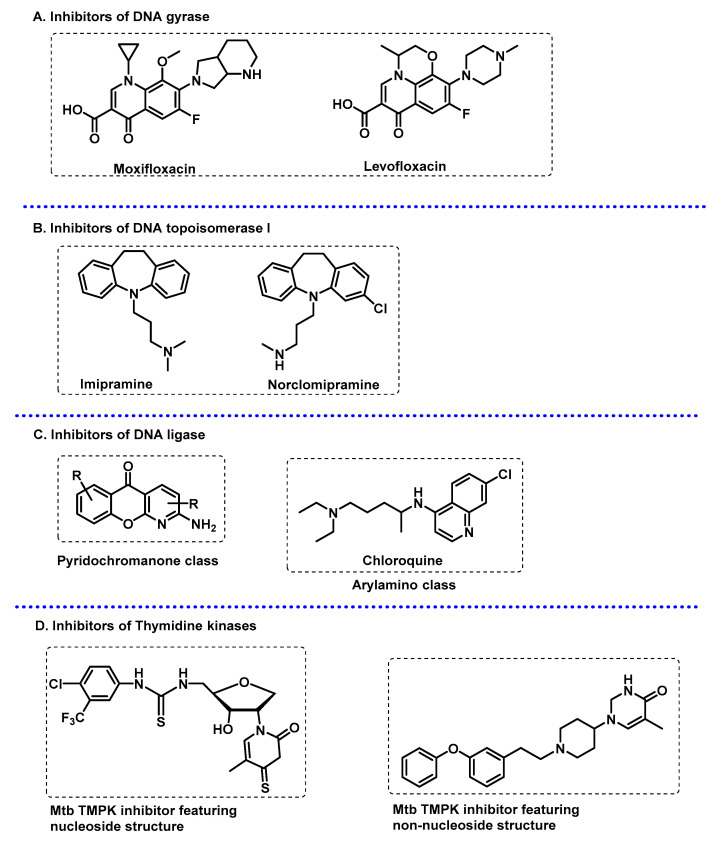
DNA-related enzymes: (**A**) second-line anti-TB fluoroquinolones as DNA gyrase inhibitors; (**B**) DNA topoisomerase I inhibitors of Mtb; (**C**) examples of inhibitors of Mtb DNA ligase; (**D**) thymidine monophosphate kinase inhibitors.

**Figure 8 ijms-24-10482-f008:**
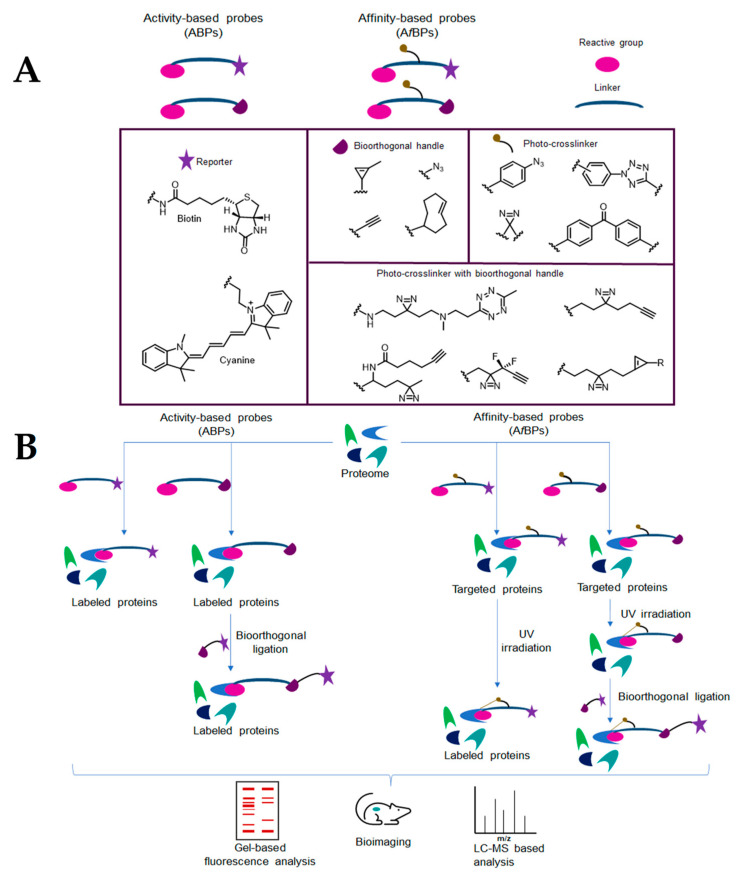
(**A**) Structure of activity-based probes and affinity-based probes. Overview of common structures used as reporters, bio-orthogonal handles, and photo-crosslinkers. (**B**) Workflow for protein profiling using activity-based probes and affinity-based probes.

**Figure 9 ijms-24-10482-f009:**
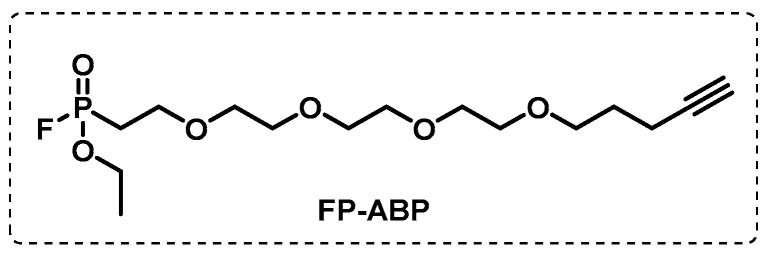
Fluorophosphonate (FP) activity-based probe.

**Figure 10 ijms-24-10482-f010:**
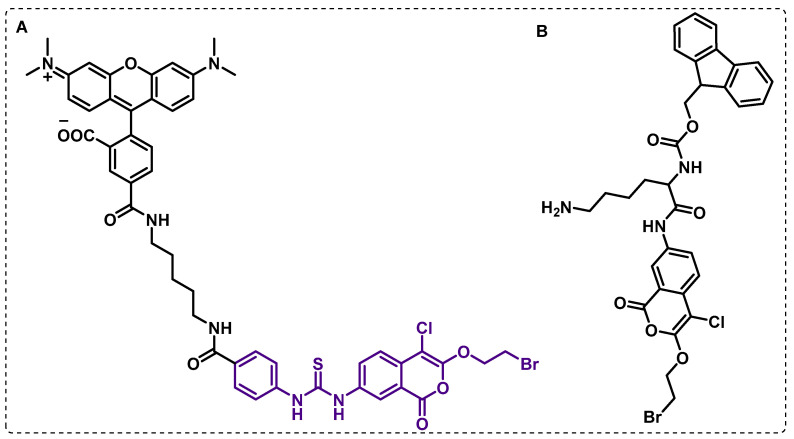
(**A**) Isocoumarin-based probe synthesised based on the inhibitor, (**B**) potent inhibitor design with nanomolar activity.

**Figure 11 ijms-24-10482-f011:**
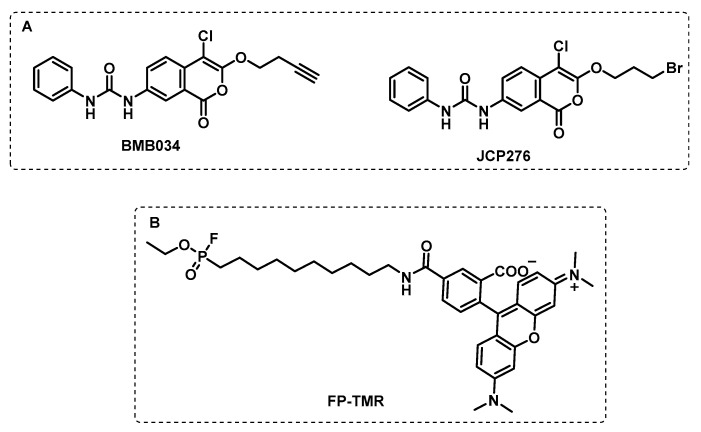
(**A**) BMB034 ABP design based on the JCP276 serine inhibitor. (**B**) Fluorophosfonate tetramethylrhodamine ABP.

**Figure 12 ijms-24-10482-f012:**
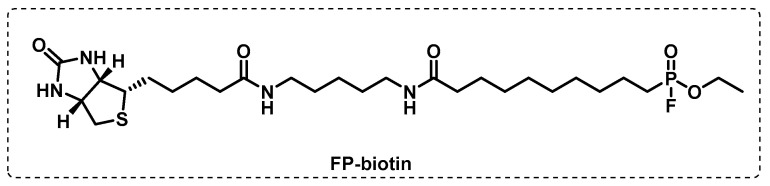
Fluorophosfonate biotin probe.

**Figure 13 ijms-24-10482-f013:**
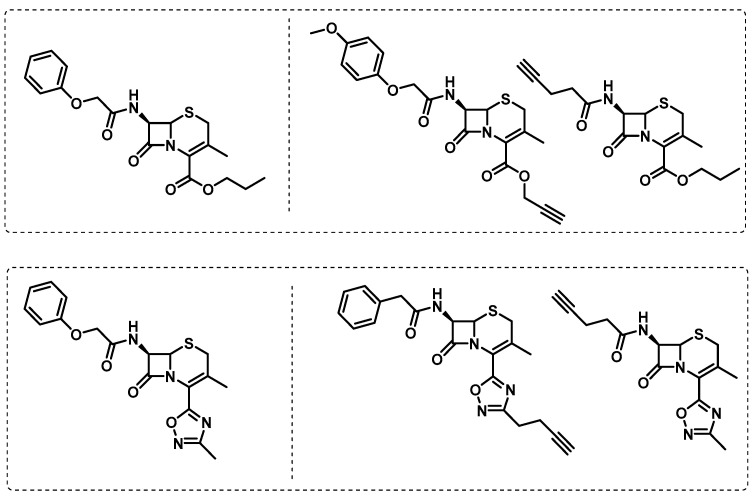
Cephalosporine alkyne analogue probes.

**Figure 14 ijms-24-10482-f014:**
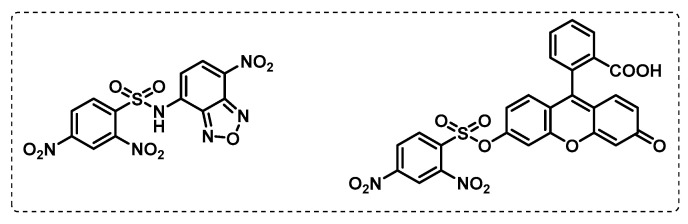
Sulfone fluorogenic probes: benzoxadiazole (**left**) and fluorescein (**right**).

**Figure 15 ijms-24-10482-f015:**
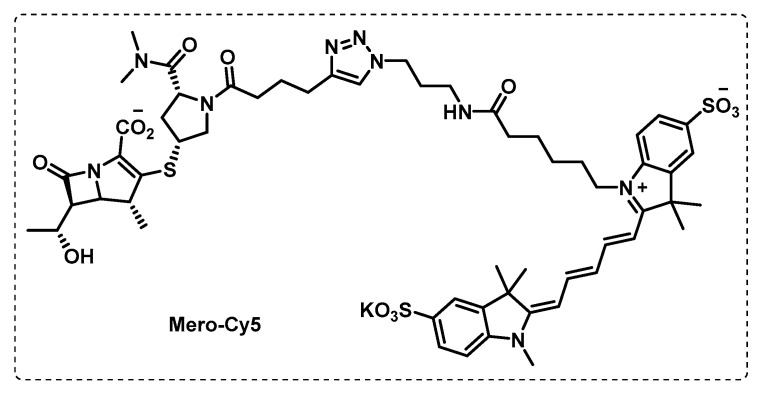
Meropenem-based ABP.

**Figure 16 ijms-24-10482-f016:**
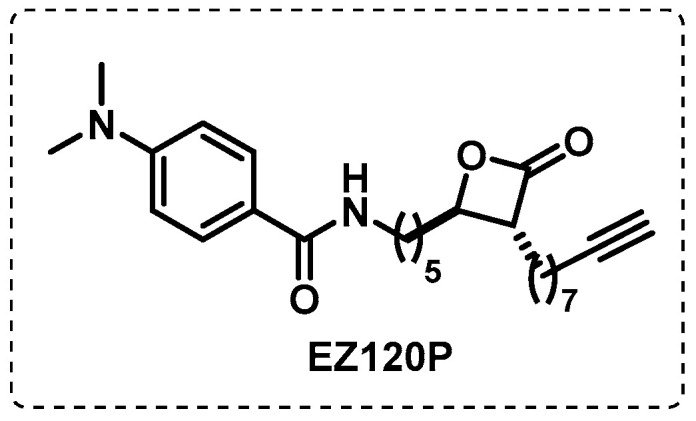
*β*-lactone-based ABP.

**Figure 17 ijms-24-10482-f017:**
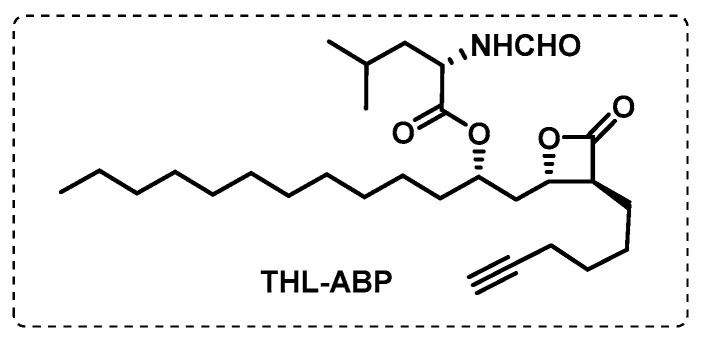
Tetrahydrolipstatin-based ABP.

**Figure 18 ijms-24-10482-f018:**
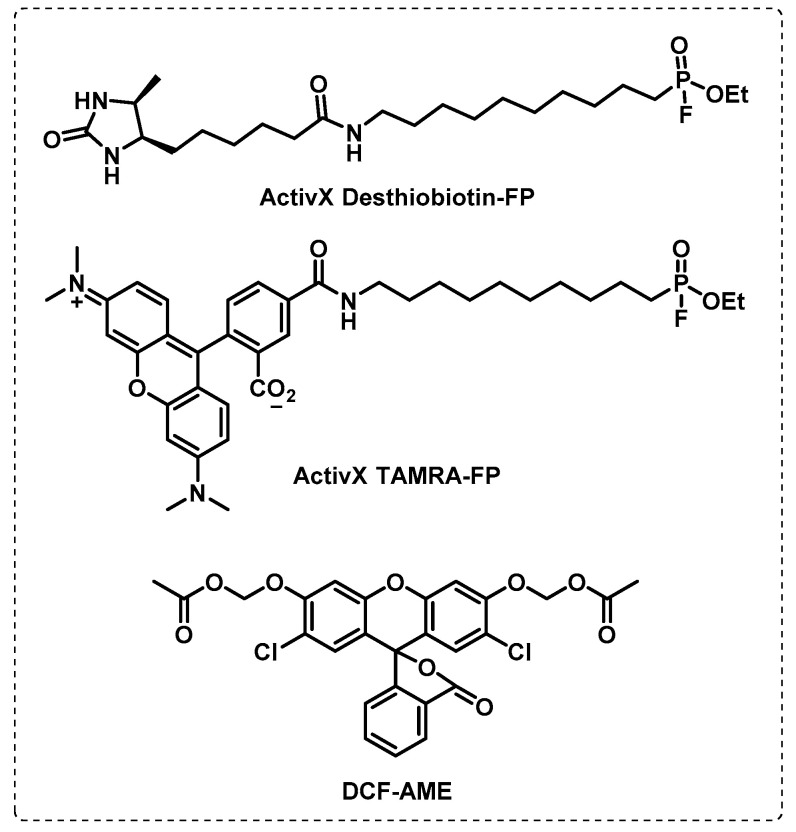
Biotin probe (ActivX Desthiobiotin-FP) and fluorogenic probes (ActivX TAMRA-FP and DCF-AME) used to identify esterases.

**Figure 19 ijms-24-10482-f019:**
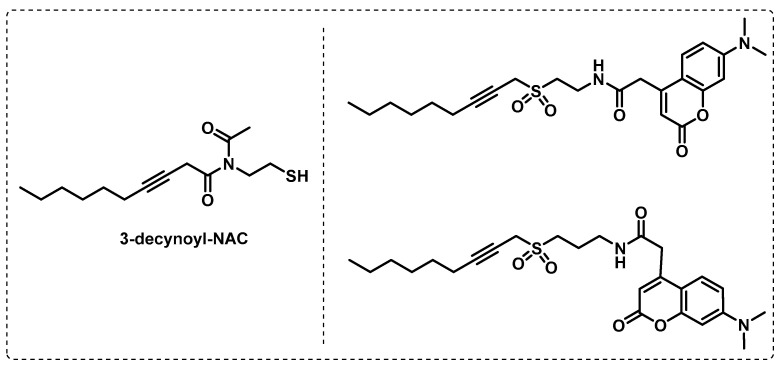
3-decynoyl-NAC-based ABPs.

**Figure 20 ijms-24-10482-f020:**
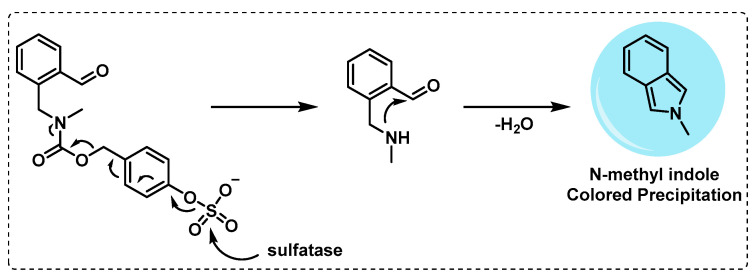
Probe used by Yoon et al. to detect sulfatase activity.

**Figure 21 ijms-24-10482-f021:**
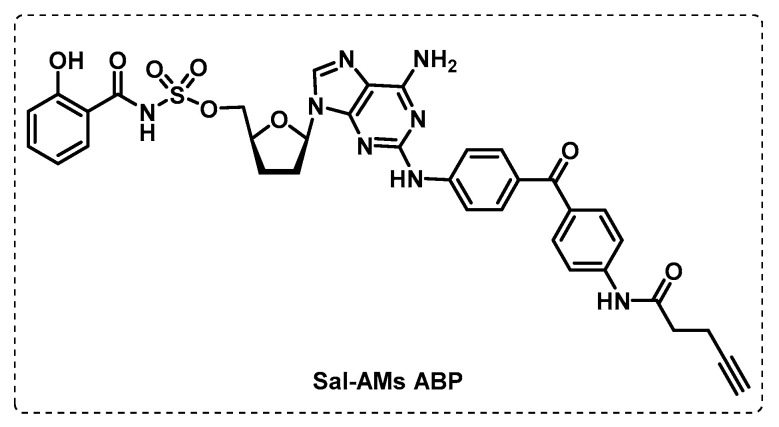
Sal-AMs ABP.

**Figure 22 ijms-24-10482-f022:**
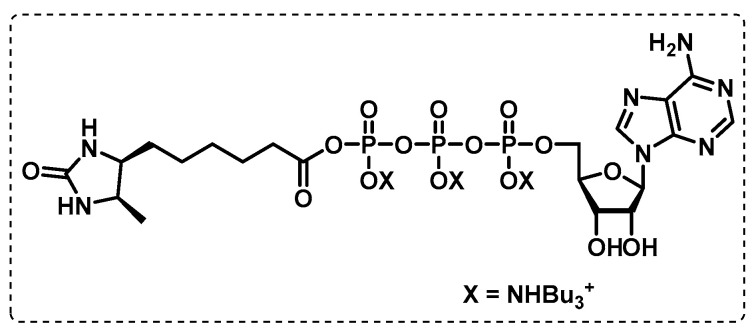
Desthiobiotin-ATP probe.

**Figure 23 ijms-24-10482-f023:**
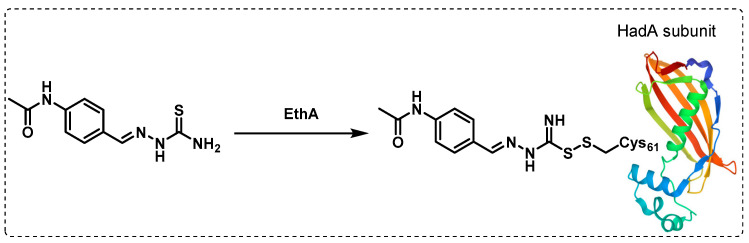
Thioacetazone is activated by monooxygenase EthA and then binds covalently to the HadA subunit of the dehydratase complex HadAB.

**Figure 24 ijms-24-10482-f024:**
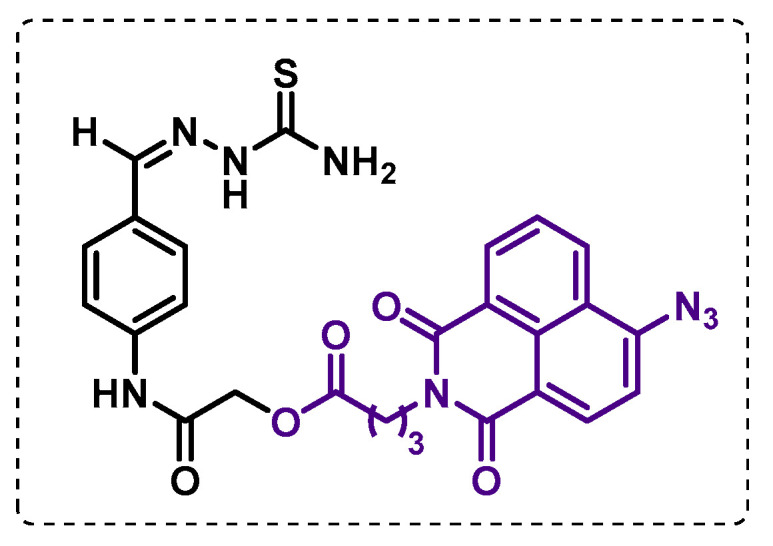
TAC-affinity-based probe.

**Figure 25 ijms-24-10482-f025:**
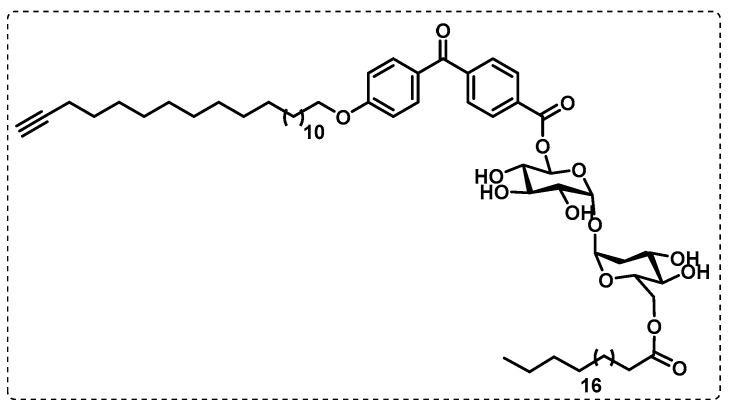
AfBPP-TDM probe.

## Data Availability

Not applicable.

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
