# Peer review of "Target Identification in Anti-Tuberculosis Drug Discovery"

_ijms, 2023, doi:10.3390/ijms241310482_

Round 1

Reviewer 1 Report

he publication is a review of current and potential therapeutic strategies in the treatment of tuberculosis. The publication presents a number of modern therapeutic regimens based on M tuberculosis protein targets. The work is very condensed in terms of content, however, the presentation of all new potential anti-TB leading structures would require a book form. In my opinion it would be interesting to briefly present the methods of designing new antituberculous drugs in the publication.
Also there is no information about the available spatial structures of druggable Mtb proteins, which are the basis for designing new drugs based on the structure of the target (SBDD).

A technical error was noted on a literature reference (line 323)

Author Response

1) The work is very condensed in terms of content, however, the presentation of all new potential anti-TB leading structures would require a book form.

We agree with the reviewer in the sense that this a highly condensed manuscript resulting from the range of topics covered. However, we believe that the way the manuscript is organized, enables the reader to understand the key challenges in TB target and drug discovery.

2) In my opinion it would be interesting to briefly present the methods of designing new antituberculous drugs in the publication.

While we understand the interest of the  reviewer to briefly present  the methods used for  drug design, we believe that this action would significantly extend the manuscript. However, we provide in the manuscript a summary of the methods used in anti-TB drug design at different parts of the manuscript, e.g.

pp2- “Currently, a diverse array of strategies is used for the development of new anti-TB therapies. The most common include genetic approaches for the identification of new molecular targets, large-scale cell-based screening trials using Mtb, virtual screening, structural biology approaches, and optimizing existent drugs through molecular modifications. Combinations between approaches based on validated targets and cell-based screening trials have been gaining attention in recent years and seem a promising strategy to discover new active drugs.”

pp2/3- “While historically effective, high-throughput screenings encounter several challenges in TB drug discovery. Despite they have general high hit rates, many compounds have undesirable physicochemical attributes, low selectivity, or mammalian cytotoxicity [17,21]. With the evolution of genomic tools, target-based screenings on validated drug targets presumed to be indispensable for the survival of Mtb and pathogenicity have gained some attention. Combination of ligand-based and structure-based chemogenomic approaches followed by biophysical and biochemical validation have also been used to identify targets for Mtb phenotypic hit.

Pp22- “The full integration of classical phenotypic screening and genomic approaches with the proteomic-based protein profiling will be instrumental to identify effective targets to develop new safer and efficacious drug candidates”.

3) Also there is no information about the available spatial structures of druggable Mtb proteins, which are the basis for designing new drugs based on the structure of the target (SBDD).

Assuming that the reviewer is referring to the pdb codes of the proteins/enzyme, we believe that adding such information would require additional text explaining the use of SBDD, thus further unnecessarily expanding the length of the manuscript

A technical error was noted on a literature reference (line 323) - point solved

Reviewer 2 Report

In this manuscript, the authors present an overview of the current challenges in TB drug discovery and emerging Mtb druggable proteins, and how chemical probes for protein profiling enabled the identification of new targets and biomarkers. The review is clearly written. The topic was described honestly. The cited references are mostly recent publications, from 2018-2022 years. The figures are appropriate and easy to interpret. The pathophysiology of TB is quite complicated and the molecular targets or biological pathways are not fully known. So, such reviews are needed. The work may be of interest to scientists. I have no significant substantive comments. Only minor remarks:

Line 323-324 “(Error! Reference source not found.Figure 4C) ??

Line 206 for the drug ethambutol. [51]” should be ….[51].

Author Response

Line 323-324 “(Error! Reference source not found.Figure 4C) ?? - point solved

Line 206 for the drug ethambutol. [51]” should be ….[51]. - point solved

Reviewer 3 Report

Dear Authors,

The theme that you selected for your review is of great interest in medicinal chemistry.

The spread of tuberculosis all over the world, the easy contamination and the emergence of high resistant mycobacterial strains require the development of new active therapies or the discovery of new drug targets.

The manuscript is well-organized, well-written. the references are well chosen.

Author Response

Not applicable